# Relative Frequencies of *PAX6* Mutational Events in a Russian Cohort of Aniridia Patients in Comparison with the World’s Population and the Human Genome

**DOI:** 10.3390/ijms23126690

**Published:** 2022-06-15

**Authors:** Tatyana A. Vasilyeva, Andrey V. Marakhonov, Sergey I. Kutsev, Rena A. Zinchenko

**Affiliations:** 1Research Centre for Medical Genetics, 115522 Moscow, Russia; vasilyeva_debrie@mail.ru (T.A.V.); kutsev@mail.ru (S.I.K.); renazinchenko@mail.ru (R.A.Z.); 2N.A. Semashko National Research Institute of Public Health, 105064 Moscow, Russia

**Keywords:** 11p13, *PAX6*, de novo mutation, chromosome deletions, small indels, aniridia

## Abstract

Genome-wide sequencing metadata allows researchers to infer bias in the relative frequencies of mutational events and to predict putative mutagenic models. In addition, much less data could be useful in the evaluation of the mutational frequency spectrum and the prevalent local mutagenic process. Here we analyzed the *PAX6* gene locus for mutational spectra obtained in our own and previous studies and compared them with data on other genes as well as the whole human genome. MLPA and Sanger sequencing were used for mutation searching in a cohort of 199 index patients from Russia with aniridia and aniridia-related phenotypes. The relative frequencies of different categories of *PAX6* mutations were consistent with those previously reported by other researchers. The ratio between substitutions, small indels, and chromosome deletions in the 11p13 locus was within the interval previously published for 20 disease associated genomic loci, but corresponded to a higher end due to very high frequencies of small indels and chromosome deletions. The ratio between substitutions, small indels, and chromosome deletions for disease associated genes, including the *PAX6* gene as well as the share of *PAX6* missense mutations, differed considerably from those typical for the whole genome.

## 1. Introduction

Different genomic loci may be subject to specific mutagenic mechanisms. Replication errors and/or DNA damage leave their own footprints in the sequence leading to different spectra of mutations occurring at different genomic loci [1]. Heterozygous *PAX6* pathogenic variants or chromosome 11p13 rearrangements cause congenital aniridia, related phenotypes, and, on rare occasions, other phenotypes such as isolated optic nerve malformations, foveal hypoplasia, and achromatopsia. The main underlying mechanism consists in the haploinsufficiency of *PAX6* function. On the one hand, the very first *PAX6* researchers pointed to specificities of 11p13 genome locus mutational events frequencies [2]. On the other hand, the spectra of *PAX6* gene mutational events in different cohorts have recently become more explored, though the degree of their similarity and the relative frequencies of mutations remain unclear. In both of these respects, a comparative study of *PAX6* mutations spectrum and its consideration in the world population context, as well as in the context of the whole genome, could be useful, as it could help to elucidate spectrum peculiarities, which could point to putative mechanisms of local 11p13 mutations.

## 2. Results

### 2.1. PAX6 Mutation Type Proportions and Their Consistency with Mutational Spectra from Three Large Studies of Aniridia Patients

The current study revealed 138 disease-causing variants in the *PAX6* gene and 61 chromosome 11p13 rearrangements. Defined chromosome deletions varied from 0.35 to 7.5 Mb in size. The 138 small intragenic *PAX6* variants included 53 nonsense pathogenic variants, 30 splicing changes, 34 frame shifts, 8 CTE (C-terminal extension), 2 start codon losses, 7 missense pathogenic variants, and 4 pathogenic variants in 5′-UTR sequence (Appendix A) [3,4,5,6].

The *PAX6* mutations ratio obtained in the current study was characterized by high proportions of de novo mutations in the sample (147 sporadic cases out of 199, or 73.9%) as well as of chromosome deletions (61 out of 199, or 30.7%), frequent 3′-*cis*-regulatory region deletions (15 out of 199, or 7.5%), and a rarity of nonsynonymous substitutions (7 out of 199, or 3.5%), while the great majority of sequence *PAX6* variants were premature stop codon formation mutations (117 out of 199, or 58.8%).

The comparison of *PAX6* mutation types was carried out for (i) four studies with the number of examined families more than 50 (from Great Britain [7], the USA [8], China [9], and Russia (current study)) as well as for (ii) five other studies with smaller numbers of patients (13 < N < 30) (Table 1). The analysis demonstrated a common pattern of *PAX6* mutation types in different populations around the world, even considering small cohorts (Kruskal–Wallis test, *H* = 15.05, approximate *p*-value = 0.0582).

Highlighting the similarities and differences between the cohorts, the proportion of 11p13 chromosome deletions presumably was lower in other studies, but, in some cases, it could be higher (30/77, or 39% [10]); the same was true for the fraction of 3′-*cis*-regulatory region deletion which could account for 11/141, or 7.8% [11]. An extremely high frequency of indels in the *PAX6* gene was observed in every study and seemed to be typical. That last observation was also noted by Bobilev et al. in their *PAX6* mutation study [8]. Very few missense pathogenic variants were identified in seven Russian families with congenital aniridia or related phenotypes (7/199, 3.5%),but a similar fraction was also typical and previously observed (6/84, 7.1%) [2].

### 2.2. Relative Frequencies of PAX6 Mutations vs. Data on Other AD Diseases Associated Genes

To compare data from the current study with the pattern of spontaneous mutations in other genes and the whole human genome, we counted all our variants as de novo events, which was permissible for congenital aniridia since even familial mutations arose as de novo. For the Russian cohort, we have 91 single nucleotide substitutions, 47 indels of <50 bp in length, and 61 de novo chromosome events. 

Firstly, we compared the *PAX6* ratio obtained in the current study with data for 20 genes associated with dominantly inherited fully penetrant diseases reviewed by Kondrashov [17]. Mutations in twenty diseases associated genes were defined by targeted sequencing and copy number variation analysis as they were in our study.

We received a higher proportion between substitutions and small intragenic indels (less than 50 bp), at a ratio of 91:47 (~1.9:1). The ratio could vary across different gene loci, but the proportion 1.9:1 was close to the highest edge of the interval published by Kondrashov. The ratio among 20 loci varied from 0.6:1 for the *APC* gene to 15.6:1 for the *F9* gene [17]. While comparing the incidence of small indels in two well characterized genes, *PAX6* and *F9*, Prosser and van Heyningen also noted that in the *PAX6* locus they occurred several times more frequently than in the *F9* [2]. This observation appeared not to be a result of sampling bias, as was suggested by Proser and van Heyningen [2], since the *PAX6* gene locus usually showed the same high proportion of small indels (Table 1) and the *PAX6* and *F9* genes were located at the opposite sides of distribution of the proportions (Table 2).

The occurrence of 11p13 chromosome large rearrangements seemed to be too frequent in our study as well as in other *PAX6* studies, and evidently was typical for the locus (Table 1). In our cohort, 11p13 chromosome deletions were responsible for 30.7% of the aniridia cases. The ratio between substitutions and chromosome events counted for 91:61 (1.5:1) in the current study. Compared with 20 disease-associated genes in Kondrashov’s review, the share of 11p13 deletions was higher than for almost all reviewed loci; and it was as large as for the *VHL* and *APC* genes (1.6:1 and 2.1:1, respectively), and less only than *F8* and *DMD* genes (0.7:1 and 0.5:1, respectively) (Table 2) [17]. Furthermore, according to our data, 11p13 CNVs occurred more frequently (the ratio between substitutions and CNVs was 1.5:1) than it was estimated in the Kondrashov’s review (7.4:1). The discrepancy could be related to the improvement of the molecular diagnosis methods and routine usage of CNVs analysis in the modern aniridia DNA diagnosis protocols.

### 2.3. Relative Frequencies of PAX6 Mutations vs. the Whole Genome De Novo Mutations

To assess relative frequencies of *PAX6* mutations in the background of the whole genome, we compared the ratio between substitutions, micro indels, and large chromosome events revealed in this study with that obtained a sequencing project for the whole genome in 250 Dutch families (Table 3) [18,19]. The comparison demonstrated a distinguished *PAX6* mutation pattern with elevated proportions of small indels and chromosome deletions. The ratio between de novo substitutions, indels, and large deletions in the current study was 91:47:61 (1.5:0.8:1), and differed from the whole genome proportion of 11020:291:41 (268.8:7.1:1) (Freeman–Halton extension of Fisher’s exact test *p*-value = 0.0036).

Indeed, defined in healthy samples, de novo large CNVs seemed to be relatively rare compared to indels and SNVs (one de novo CNV per 7 births occurs, while on average 9 de novo indels and 73 de novo SNVs per birth occur) [20].

As mentioned above in our study, *PAX6* missense variants were identified in few probands (7/199, or 3.5%), and such a proportion appeared to be typical for the *PAX6* gene. We tried to infer whether the fraction of missense variants in the *PAX6* gene indeed distinguished the locus from the whole human genome. 

Previously, Hanson et al. pointed to the highly biased *PAX6* mutations spectrum in aniridia patients with only 2% of missense substitutions, as the missense rarity was proposed to be one more feature of the locus [21]. In a retrospective review of *PAX6* molecular diagnosis in a large cohort of patients with several different clinical diagnoses, the share of missense pathogenic variants could reach 12.2% (31/254 [11]). Thus, *PAX6* missense pathogenic variants could be observed in extended samples of the patients, although in a much lower proportion than expected based on the whole genome data. In the whole genome databases of 10 930 small-scale de novo mutations for monogenic disorders, 37% of mutations were missense [22]. The ratio between de novo nonsynonymous and nonsense pathogenic variants in the coding region established in the 78 Icelandic Trios sequencing project was ~30:1 [23]. While even considering all known *PAX6* missense variants, including benign variants met in healthy samples (and registered in gnomAD datasets) and pathogenic missense variants identified in affected people (gathered in the HGMD dataset), the proportion between *PAX6* nonsynonymous and nonsense substitutions would be only ~2:1 (367:179) (https://gnomad.broadinstitute.org/ (accessed on 14 May 2021) and http://www.hgmd.cf.ac.uk/ (accessed on 14 May 2021)). The number 367 is formed by 133 known *PAX6* missense pathogenic variants from HGMD and 234 ones known from gnomAD; the number 179 is formed by 79 nonsense pathogenic variants from HGMD database plus at least 100 different cases of *PAX6* hot spot nonsense mutations (20% of all known *PAX6* mutations according to Tzoulaki’s statistics), counting every case of recurrent nonsense mutation in the hotspots as a mutation [24]. The proportion ~2:1 between *PAX6* nonsynonymous and nonsense substitutions demonstrated an extremely lower share of *PAX6* missense mutations. 

The rarity of missense mutations in the *PAX6* gene is possibly explained by strong negative selection pressure towards them, as some *PAX6* nonsynonymous substitutions could cause severe consequences. It has been proposed that they have a dominant negative effect and lead to severe or nonviable phenotypes [25].

### 2.4. Comparison of the Sequence Context of PAX6 Mutations with Data of the PAX6 Mutational Review by Prosser and Van Heyningen

Next, we tried to assess the sequence context of *PAX6* substitutions. To explore the contexts of pathogenic *PAX6* variants and to characterize the mutation process in the cohort, we compared our study with previous ones. In 1998, Prosser and van Heyningen [2] reviewed the spectrum and sequence context of known *PAX6* pathogenic variants (N = 84) and discussed their putative mechanisms. In our cohort, we have a comparable number of intragenic *PAX6* pathogenic variants (N = 138) (Table 4). A comparison of mutational spectrum in our cohort of patients with those analyzed by Prosser and van Heyningen revealed no significant difference (Mann–Whitney *U* = 10, exact *p*-value = 0.2403). Thus, benchmarking could be relevant.

Proser published a 24:16 ratio of transitions at CpG to non-CpG dinucleotides, while we reported 31:20. Therefore, in both studies it was about 3:2 [2].

In our study, C-to-T substitutions occurred in 6 CpG *PAX6* sites, in codons 44 (once), 103 (3 times), 203 (8 times), 240 (10 times), 261 (2 times), and 317 (3 times). In addition, the difference with Prosser’s analysis accounted for two points. We did not meet changes in CpG in 128 and 208 positions, but met them in two other ones: in 44 (once) and in known hotspot 261 (twice).

These eight CpG sites are highly mutable and present de novo *PAX6* mutation hot spots in every study in different populations. All in all, there are 44 CpG dinucleotides in the coding sequence of the *PAX6* gene [2]. Prosser proposed that mutation hotspots should be sites of methylation. They could not, at least, be the positions of these eight hot spots not constantly methylated (UCSC methylation map, https://genome.ucsc.edu/ (accessed on 14 May 2021)). It is currently generally accepted that methylation on its own does not entirely explain the mutability of CpG sites [26].

Other hot spot mutations identified in our cohort were localized at positions c.109, c.265, c.300, c.357+1, c.467, c.916+1, c.1183, c.1183+2, and c.1268. Probands from 18 unrelated families were found to have known changes at those points.

Hot spot mutations defined in the Russian cohort counted for 24% of all mutations (48/199). That corresponded with Kondrashov’s conclusion that mutations in hot spots (those sites where independent mutations were observed three or more times were regarded as hot spots) might make only a minor contribution to spontaneous substitutions [17], though the share of 24% did not seem literally “minor.” Recurrent mutations could explain why all *PAX6* mutations arose in the same proportions in different populations, according to same rules possibly dictated by local 11p13 chromosome context and architecture.

### 2.5. Comparison of the Sequence Context of PAX6 Mutations with the Whole Genome Data

We assessed the differences in frequencies of context-specific mutational events in the *PAX6* locus and in the whole genome. The ratio between *PAX6* transitions and transversions obtained here, 61:31 (2:1), was in consistent with a reported ratio for the whole genome [27]. Transitions to a transversions ratio appeared to be 2:1 across the whole genome, 3:1 in the coding part of the genes, and 2:1 for nonsynonymous substitutions in the coding sequence [28]. The ratio between de novo transitions and transversions was defined as 2:1, both in the UK10K Project and in the 78 Islandic Trio analysis [23,29].

The ratio of transitions at CpG to non-CpG dinucleotides as defined for the *PAX6* coding sequence in the Russian cohort was 31:20. Similar transition distribution between CpG and non-CpG dinucleotides could be observed among nonsense pathogenic variants in 20 disease-associated loci [17]. The ratio between the number of C-to-T substitutions (as well as complementary G-to-A substitutions) to the number of changes in the opposite direction (T-to-C or A-to-G) obtained here was 25:12 (2.1:1), which was consistent with a previously published ratio (2.15:1) The ratio was established based on direct approaches to genome wide de novo substitution estimations in families [23].

It was interesting to compare the ratio of novel and recurrent mutations for the *PAX6* locus and for the whole genome. In our study, recurrent *PAX6* sequence variants were defined in 75 probands, while novel were defined in 63 probands (the ratio was approximately 1.2:1). That ratio was also typical for other *PAX6* studies (61:70, ≈1:1.1 [11]), as well as for the novelty rate of the whole genome mutations 1:1 [30].

## 3. Discussion

A large cohort of 199 patients from Russia with aniridia or aniridia-related phenotypes was studied for the *PAX6* mutation spectrum, considering the background of the *PAX6* mutation spectrum in the world’s population and the ratio of de novo mutations in the whole genome.

We compared our results with studies of three large cohorts of aniridia patients (each with more than 50 patients): Robinson et al. [7], Bobilev et al. [8], and Bing You et al. [9], who screened their patients by searching for the *PAX6* gene point pathogenic variants and 11p13 chromosome deletions. The similarity of methodology considering the high frequency of 11p13 chromosome events allowed for a comparison of results obtained in all three studies. Robinson et al. [7] analyzed a number of patients with both syndromic aniridia and WAGR syndrome. Based on our experience, WAGR patients are rarer in the cohorts of patients with isolated aniridia, as they receive karyotyping diagnostics very early. That possibly explains a higher share of chromosome events in Robinson’s study, although the discrepancy was not relevant.

Then we compared our data with that obtained earlier for other disease-associated loci. A comparison with Kondrashov’s study [17] of rates of different types of mutations in 20 disease-associated loci was acceptable, as in both analyses data were obtained (i) for loci where loss-of-function mutations produce clear-cut phenotypes, (ii) about 100 independent patients with mutations in the locus were analyzed, and (iii) a fraction of patients with no pathogenic mutations detected at the locus was well below ½. Kondrashov also considered only loci associated with diseases with dominant inheritance, full penetrance, and haploinsufficiency as the pathogenic mechanism, for which all coding sequence and chromosome events were routinely analyzed [17].

In addition, for the comparison with the whole genome data, we gathered relative frequencies of different types of mutations estimated in two analyses. Both analyses were based on the same project of the whole genome NGS sequencing of 250 Dutch parent offspring families [18,19]. One analyzed de novo SNVs and indels, while the other analyzed structure variants.

In our study, we defined 138 *PAX6* intragenic variants, including 53 nonsense mutations, 30 changes of splicing, 34 frame shifts, 8 CTE, 2 start codon losses, 7 missense mutations, and 4 pathogenic variants in 5′-UTR sequence and 61 chromosome 11p13 rearrangements. 

Despite the high share of de novo mutations, which were four times more frequent than inherited ones, the *PAX6* mutational events relative frequencies in Russian aniridia patients’ cohort remained the same as those obtained in earlier studies of different populations. The similarity between different populations might provide an insight into the common mechanism of *PAX6* locus mutations constantly arising at a specific rate for each type of mutation. The resolving condition and perhaps direct reason for this might be specific base composition, repeat content, chromatin state and structure, nucleosome position, methylation, and other epigenetics marks, as well as repair efficiency and other specific related properties of this genome locus [1,31]. The contribution of the hot spot events in the *PAX6* mutation spectrum (which was defined as 24%) seemed not to be so modest as was previously established by Kondrashov for other 20 disease-associated genes [17]. Peculiar for the *PAX6* locus mutations spectrum is a higher proportion between substitutions and chromosome deletions (91:61, approximately 1.5:1). The frequency could be typical for the locus, as 11p13 deletions appeared to be characterized by some common features. Previously, we showed that the 11p13 locus had a propensity to be de novo broken, presumably on the paternal 11p13 chromosome [32]. Another peculiarity among the *PAX6* gene features is a higher proportion between substitutions and small (<50 bp) indels, which was defined as 91:47 (approx. 2:1). Ratios obtained here were within the interval reported by Kondrashov for 20 disease-associated loci, but corresponded to its higher end. The rarity of the *PAX6* gene missense mutations accounts for one more spectrum peculiarity. We could observe “levelling out the differences” between *PAX6* mutations types. Several possible factors could play role, such as high conservativeness of the *PAX6*, the deleterious effect of most *PAX6* substitutions, and higher frequencies of the indels and CNVs typical for the region.

On the putative mechanism of the *PAX6* gene mutations formation, Prosser has suggested DNA damage is influenced by environment factors as well as endogenous processes. But a collation of the data of different population seems to demonstrate no differences in the *PAX6* mutational spectra, which implies that *PAX6* mutations arise through the same mechanisms. Indeed, endogenous processes as a principal factor lead to the same rates of different kinds of mutations in the same locus. The rates strongly correlate inside numerous genome loci depending on, e.g., sequence context and recombination rate [33]. The relative frequency of high *PAX6* indels has been previously reported [8]. Along the human genome, small indels frequency varies widely. About 40% of small indels occur in 4% of the genome, and they arise 16 times rarer than SNPs in the remaining 96% of the genome sequence [34]. The frequency of indels observed here in the *PAX6* locus could be explained, e.g., by DNA strand slippage and/or breaks in sites of high chromatin rigidity due to low local cleavage intensity, as indels distribution was shown to be strongly associated with low DNA cleavage intensity [35]. Moreover, the same chromatin features could correspond to the breaks’ formation and a high frequency of large chromosome rearrangements in the 11p13 locus [35].

Interestingly, the proportions between SNVs, indels, and CNVs for 20 disease-associated loci, including the *PAX6* gene, differed considerably from de novo genome-wide mutations shares. Evidently such a shift of proportions between de novo SNVs, indels, and CNVs from 44.3:11.4:1 [1] for the whole genome to 5.1:1.3:1 (total sums) for 20 disease-associated loci [17], and to 1.5:0.8:1 for 11p13 locus obtained here, could be explained by a conservativeness of the disease-associated coding sequence and selection impact [36]. Functional consequences of frame shifting indels are profoundly serious, and they obviously are strongly influenced by purifying selection [34]. Selection pressure towards substitutions along the genome could vary and be weaker, possibly explaining a lower portion of indels among the genome (11.4 ÷ 16:1) [1,34]. The lower ratio between indels and substitutions indicates less selection pressure on substitutions. That proportion is changing for conservative protein coding sequences. Considering *PAX6* along with the conservativeness of other disease-associated genes and the deleterious effect of substitutions, a shift in the proportion between SNVs and indels for disease associated loci seems to be expected. However, the overrepresentation of indels among disease-causative variants was previously shown [37]. In addition, the proportion could vary depending on the mutation rate variance in functionally different genome loci. Therefore, the de novo mutation rate is proposed to be higher in functionally active regions, in coding and regulatory ones, partially due to abundant CpG content [1,38].

Although we obtained a similar characteristic proportion between transitions and transversions in the *PAX6* locus and in a healthy genome, we need to keep in mind that we considered a sample of affected patients, and that might and should influence the proportions between sequence lesion types. Accordingly, the comparison of proportions of different lesion types with other disease-associated genes was undertaken. 

## 4. Materials and Methods

The sample consisted of patients with aniridia clinical features and identified molecular genetic cause of the phenotype, which was either 11p13 deletion or intragenic pathogenic/likely pathogenic *PAX6* variant. All such patients were referred to the Research Centre for Molecular Genetic for genetic testing. 

A total of 248 patients from 199 unrelated families were included in the study. Of these, 118 (47.6%) are male and 130 (52.4%) are female. The mean age at examination was 16.8, median 9.0 years (25%–75%, range 3–28, minimum 0.2, maximum 65.0). Of the 199 probands, 147 (73.9%) were sporadic and 52 (26.1%) had a family history. Of the 13 patients were diagnosed with the WAGR syndrome, 1 was diagnosed with WAGRO, 1 family was diagnosed with achromatopsia, 1 more was diagnosed with optic nerve hypoplasia, and the remaining 183 families received a diagnosis of congenital aniridia. The patients were examined by ophthalmologists from four clinics, the Research Centre for Medical Genetics, Moscow; the Cheboksary Branch of the S. Fyodorov Eye Microsurgery Federal State Institution; the Moscow Helmholtz Research Institute of Eye Diseases; and the National Medical Research Centre for Children’s Health, Moscow. 

The majority of the patients (181/248, or 73%) considered themselves to be Russian, while the rest of the patients considered themselves as belonging to several nationalities: Tatar (17/248, 6.9%), Ukrainian (6/248, 2.4%), Chuvash (7/248, 2.8%), Kazakh (6/248, 2.4%), Kirgiz (6/248, 2.4%), Belarusian (N = 3), Moldovan (N = 3), Ashkenazi (N = 3), Chechen (N = 4), Udmurts (N = 3), Armenian (N = 2), Avar (N = 2), Dargin (N = 2), Karachai (N = 2), and Bulgarian (N = 1).

Screening for the 11p13 rearrangements was carried out by Multiplex Ligation-dependent Probe Amplification (MLPA analysis) by using SALSA MLPA Probemix P219 according to manufacture protocol (www.mlpa.com) Screening for pathogenic variants in the *PAX6* gene was carried out by a Sanger sequencing analysis of 12 *PAX6* exons (3–13) and their exon-intron boundaries. 

Chromosome 11p13 deletions were designated according to the MLPA probes with reduced signal coordinates according to the NCBI36/hg18 assembly of the human genome. Intragenic pathogenic variants were named based on *PAX6* transcript variant 1 (NM_000280.4) according to the HGVS nomenclature. The pathogenic status of single nucleotide variation for interpretation of sequence variants was established using the consensus recommendations of the American College of Medical Genetics and Genomics and the Association for Molecular Pathology [39]. Only pathogenic and likely pathogenic genetic changes were reviewed.

Mutations identified in 139 index patients were described in our previous publications [3,4,5,6], while for 60 families. 46 mutations were newly reported. They were 31 *PAX6* intragenic mutations identified in 36 families and 15 different deletions defined in 24 families. Mutations of new index patients are listed in Appendix A. In total, the proportion of familial to sporadic probands counted for 52/147. DNA analysis was carried out by using a combination of multiplex ligation-dependent probe amplification (MLPA) analysis for searching for copy number variations (CNVs) in 11p13 chromosome region, followed by Sanger sequencing for *PAX6* intragenic variants searching [3]. 

We compared our proportions with those observed in three large studies of aniridia patients from the USA, Great Britain, and China [7,8,9]. We also included five smaller cohorts in the comparative analysis (from France [13], Australia [14], Korea [15], India [16], and another smaller cohort from Great Britain [12]). To compare the 11p13 locus with other genome loci we used data for 20 disease-associated genes reviewed by Kondrashov [17]. Kondrashov analyzed only data for diseases with dominant inheritance, nearly full penetrance, and haploinsufficiency as molecular mechanisms of pathogenesis, so it was of relevance to compare *PAX6* spectrum with those 20 reviewed loci. The spectrum of genome-wide de novo mutations was taken from a sequencing project of 250 Dutch families for the whole genome [18,19]. Kruskal–Wallis *H* test, Freeman–Halton’s extension of Fisher’s exact test, and the Mann–Whitney *U* test were used in GraphPad Prism 8.0.1 (GraphPad Software, CA, USA) to compare proportions of different types of *PAX6* mutations among several cohort studies, as well as with estimation of relative frequencies of de novo mutational events in human genome based on the data from the sequencing projects listed above.

## 5. Conclusions

Compared to worldwide aniridia patients’ samples, the relative frequencies of *PAX6* mutations obtained in our study retained the same rate, and could characterize *PAX6* spectrum. Compared to the whole genome, the relative frequencies or de novo mutations typical for the *PAX6* mutational spectrum stands out from the typical for the healthy genome, but is part of the interval for other disease-associated genes spectra. This is the case, firstly, due to shifting ratios between the mutations of different types in disease-associated genes, and, secondly, due to intrinsic peculiarities of the *PAX6* locus with a higher frequency of small indels and chromosome deletions and a lower frequency of missense mutations. 

## Figures and Tables

**Table 1 ijms-23-06690-t001:** Comparison of *PAX6* mutation types in world-wide cohorts of aniridia patients.

Title 1	Russia 2022 (This Study)	UK 2008 [7]	USA 2016 [8]	China 2020 [9]	UK 2009 [12]	France 2003 [13]	Australia 2018 [14]	South Korea 2012 [15]	India 2015 [16]
Large chromosome deletions	61	34	10	11	1	2	6	–	–
Nonsense	53	13	18	12	4	14	5	9	1
Frame shift	34	8	16	16	4	9	2	3	7
Missense	7	–	–	2	4	1	1	1	1
Splicing	30	8	14	13	5	4	1	3	3
Start codon loss	2	–	–	–	–	–	1	–	–
CTE	8	4	3	4	4	–	1	–	1
5′UTR	4	–	–	3	–	–	–	–	–
In frame del	–	–	–	2	–	–	–	–	–
**Total**	**199**	**67**	**61**	**63**	**22**	**30**	**17**	**16**	**13**

**Table 2 ijms-23-06690-t002:** The ratio of mutation events from Kondrashov’s review and the current study (highlighted in bold).

Locus	Number of All Mutations	Substitutions/ Small Indels Ratio	Substitutions/ Major Deletions Ratio
*F9*	2018	14.7	18
*F8*	1010	2.3	0.7
*APC*	858	0.6	2.1
*VHL*	617	3.6	1.6
*CYBB*	508	3	6.4
*NF1*	368	1.6	7.2
*TSC2*	350	2.2	3.5
*ABCD1*	336	3.6	13.8
*RB1*	280	1.9	4.3
*AR*	269	7.5	16.1
*IDS*	266	3.4	3.5
*AVPR2*	250	2.7	19.6
*JAG1*	249	1	7.1
*IL2RG*	224	3.5	42.8
** *PAX6* ** **(current study)**	**199**	**1.9**	**1.5**
*BTK*	195	3.1	12.6
*EMD*	191	1.2	5
*PAX6*	176	2.9	7.4
*OTC*	160	10.8	23.5
*DMD*	123	3.4	0.5
*PKD1*	82	2.4	18.7

**Table 3 ijms-23-06690-t003:** Comparison of proportions between SNVs, indels, and large chromosome deletions observed in the *PAX6* gene in Russian cohort of aniridia patients with the whole genome ratio.

Mutation Type	Novel Mutations in This Analysis	Novel Mutations in 250 Dutch Families [18,19]
All substitutions	91	11020
Small indels	47	291
CNVs	61	41

**Table 4 ijms-23-06690-t004:** Comparison of numbers of *PAX6* substitutions in different contexts identified in two studies.

Mutation Type	Prosser J. and Van Heyningen V. [2]	%	This Analysis	**%**
Transitions at CpG	24	28.6	31	15.6
Transitions at non-CpG	16	19.0	29	15.1
Transversions at CpG	3	3.6	5	2.5
Transversions at non-CpG	13	15.5	26	13.1
All substitutions	56	67.0	91	46.2
Small indels (<50 bp)	27	32.1	47	23.1
**Total**	**84**	**100**	**199**	**100**

## Data Availability

The datasets used and/or analyzed during the current study are available from the corresponding author on reasonable request.

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
