# Peer review of "Relative Frequencies of PAX6 Mutational Events in a Russian Cohort of Aniridia Patients in Comparison with the World’s Population and the Human Genome"

_ijms, 2022, doi:10.3390/ijms23126690_

Round 1

Reviewer 1 Report

The paper review is unclear and it´s specially complicated to follow the analysis provided.

There is a large set of data and ratio comparisons and it is recommended to include summary tables. 

Author Response

First of all, we eager to thank the Reviewer 1 for detailed and thorough analysis of our manuscript as well as valuable comments and recommendations to improve it. Thereunder you could find responses to each comment.

Q1.1: The paper review is unclear and it´s specially complicated to follow the analysis provided.

There is a large set of data and ratio comparisons and it is recommended to include summary tables.

A1.1: Thank you. Right you are, the analysis is complicated with a big set of data of very different origin. That was one of the biggest difficulties of this analysis, which was not a review at the very beginning. We only tried to perform out results obtained for our large cohort of aniridia patients at the background of other similar studies and data for frequency of de novo mutational events in the whole healthy genome. But, to make a statistically significant inference, we were forced to incorporate all original data in the study. Three basic tables being rationale for final inferences are already included into the manuscript. They are Tables 1, 2 and 3. The tables present original numbers which are discussed in the text as proportions which are either differ or not differ significantly.

Reviewer 2 Report

In this review paper (MS ID: ijms-1748557) entitled “Relative frequencies of PAX6 mutational events in Russian cohort of aniridia patients in comparison with a world population and human genome” the authors analyzed the PAX6 gene locus mutational spectra in a Russian population and compare their results with different populational studies. Beyond that, it was also compared their own data with other genes as well as on the whole human genome.

After reading carefully the MS, in my opinion, I believe that the proposal of the present review is interesting and important to the field, however; further improvements are necessary to guarantee the publication in the International Journal of Molecular Sciences.

First, the authors should expand the discussion by pointing out the differences between the methodologies used in different studies to evaluate the mutations in specific genes and/or in the whole genome. Additionally, explain how this impact (or does not) the interpretation of the data incorporated in the review.

Please include in the discussion session if there is information about the severity of symptoms and the type of gene mutation or deletion in the patients included in these and other studies incorporated in this review paper.

Third point: as in this review the authors bring original data, obtained by the MLPA and Sanger methodologies, it is recommended that the authors expand the methodological description.

In view of these considerations, I believe that in order to guarantee the publication of the MS in your journal the authors needs only to improve these minor topics.

Author Response

First of all, we eager to thank the Reviewer 2 for detailed and thorough analysis of our manuscript as well as valuable comments and recommendations to improve it. Thereunder you could find responses to each comment.

In this review paper (MS ID: ijms-1748557) entitled “Relative frequencies of PAX6 mutational events in Russian cohort of aniridia patients in comparison with a world population and human genome” the authors analyzed the PAX6 gene locus mutational spectra in a Russian population and compare their results with different populational studies. Beyond that, it was also compared their own data with other genes as well as on the whole human genome.

Q2.1: After reading carefully the MS, in my opinion, I believe that the proposal of the present review is interesting and important to the field, however; further improvements are necessary to guarantee the publication in the International Journal of Molecular Sciences.

First, the authors should expand the discussion by pointing out the differences between the methodologies used in different studies to evaluate the mutations in specific genes and/or in the whole genome. Additionally, explain how this impact (or does not) the interpretation of the data incorporated in the review.

A2.1: Thank you.

Methodology of the study certainly should impact the results, thus we incorporated in the review only studies with relevant methodology.

We compared our results with three large cohorts (each with more than 50 patients) studies, Robinson et al. (7), Bobilev et al.  (8) and Bing You et al.  (9), who screened their patients by searching for PAX6 point mutations and 11p13 chromosome deletions. The similarity of methodology considering 11p13 chromosome events high frequency makes it possible to compare results obtained in all three studies. Comparison with Kondrashov’s review is acceptable as in both analyses data were obtained (i) for loci where loss-of-function mutations produce clear-cut phenotypes, (ii) there were analyzed about 100 independent patients with mutations in the locus, (iii) a fraction of patients with no pathogenic mutations detected at the locus was well below ½. Kondrashov also considered only loci associated with diseases with dominant inheritance, full penetrance and haploinsufficiency as pathogenic mechanism, for which all coding sequence and chromosome events were routinely analyzed (17).

For the comparison with the whole genome data, we gathered relative frequencies of different types of mutations estimated in two studies. The both studies (focusing on either SNV or structural variations) were based on the same project of the whole genome sequencing of 250 Dutch parent-offspring families (18, 19).

The detailed reasoning was made in the Methods section (lines 356-365). We have included an additional mentioning of the methodology compliance in the discussion section (lines 221-244).

Q2.2: Please include in the discussion session if there is information about the severity of symptoms and the type of gene mutation or deletion in the patients included in these and other studies incorporated in this review paper.

A2.1: Thank you. Concerning the information about the severity of symptoms and the type of gene mutation or deletion in the patients included in these and other studies incorporated in the paper, we can note that, on the one hand, congenital aniridia presents a clear easily diagnosed phenotype, which nevertheless could vary widely in severity. On the other hand, the reason for the clinical polymorphysm is not completely understood, that to a large extent, could be determined by personal genetic background. Thus, while describing mutation spectrum we rely on the clinical diagnosis, but variability in phenotype severity is beyond the aim of the study. By the way, we studied and established some genotype-phenotype correlations for PAX6-associated aniridia in our earlier paper cited in the manuscript. We have added a cohort clinical description and short sample summary in the Methods chapter (lines 313-326).

Q2.3: Third point: as in this review the authors bring original data, obtained by the MLPA and Sanger methodologies, it is recommended that the authors expand the methodological description.

A2.3: Thank you, we have added extended methodological description in the Methods section (lines 335-342). In view of these considerations, I believe that in order to guarantee the publication of the MS in your journal the authors needs only to improve these minor topics.

Reviewer 3 Report

The authors describe in this manuscript a large cohort of patients with PAX6-related aniridia, and compare the type of deleterious variants with other studies in the literature. More information should be included in the manuscript about the context of this cohort: institutions involved, patients recruitment process, selection of patients for this study, patients´ancestry and whether these are WAGR/WAGRO syndrome cases.

More information should be included in the Methods section about the MLPA kit and reference genome used.

CTE should be spelled out.

Mutation should be substituted by deleterious or pathogenic/likely pathogenic variant in the main manuscript. 

The relevance of the analysis in Results 2.2 should be introduced. 

Specific revisions:

line 63: microlesions in the PAX6 gene was observed in every study and seemed to be typical. Should it be microdeletions?

line 134: While even considering 134 all known PAX6 missense mutations defined both in healthy and affected people. Are these variants deleterious but found in healthy individuals? Please, clarify exactly what is considered a mutation here.

line 149: it is unusual to put a reference in the title of a section 

line 330: acknowledgments section should be filled.

Author Response

First of all, we eager to thank the Reviewer 3 for detailed and thorough analysis of our manuscript as well as valuable comments and recommendations to improve it. Thereunder you could find responses to each comment.

The authors describe in this manuscript a large cohort of patients with PAX6-related aniridia, and compare the type of deleterious variants with other studies in the literature.

Q3.1: More information should be included in the manuscript about the context of this cohort: institutions involved, patients recruitment process, selection of patients for this study, patients´ancestry and whether these are WAGR/WAGRO syndrome cases.

A3.1: Thank you. We have extended a Sample description in the Methods section (lines 313-334).

Q3.2: More information should be included in the Methods section about the MLPA kit and reference genome used.

A3.2: Thank you, we have added extended methodological description in the Methods section (lines 335-347). Q3.3: CTE should be spelled out.

A3.3: Thank you, we have added the abbreviation explanation (C-terminal extension) in the text of the manuscript (line 45).

Q3.4: Mutation should be substituted by deleterious or pathogenic/likely pathogenic variant in the main manuscript.

A3.4: Thank you. We have substituted the term where it was possible. In this case, this term is used in the molecular biological context and refers to any DNA change, single nucleotide substitution, small indel and chromosomal event, that does not refer to the clinical consequences. All defined and analyzed in the current study SNVs, indels and CNVs were assessed according to the ACMG recommendations, only pathogenic and likely pathogenic genetic changes were reviewed. We added thе clarification into the Methods description (lines 343-347).

Q3.5: The relevance of the analysis in Results 2.2 should be introduced.

A3.5: Thank you. The relevance of the comparison with mutational spectra in 20 disease-associated loci was proved in the Methods section (lines 359-363), we have added a corresponding paragraph into the Discussion section (lines 221-244).

Specific revisions:

Q3.6: line 63: microlesions in the PAX6 gene was observed in every study and seemed to be typical. Should it be microdeletions?

A3.6: They are indels. Thank you. Corrected.

Q3.7: line 134: While even considering 134 all known PAX6 missense mutations defined both in healthy and affected people. Are these variants deleterious but found in healthy individuals? Please, clarify exactly what is considered a mutation here.

A3.7: Right you are, thank you. They are pathogenic missense variants from HGMD dataset and benign from gnomAD. We have added clarification into the text.

Q3.8: line 149: it is unusual to put a reference in the title of a section

A3.8: Thank you. Corrected.

Q3.9: line 330: acknowledgments section should be filled.

A3.9: Thank you. Corrected.